# Analysis and Reliability of Anthropometric Measurements during Pregnancy: A Prospective Cohort Study in 208 Pregnant Women

**DOI:** 10.3390/jcm10173933

**Published:** 2021-08-31

**Authors:** Inmaculada Gómez-Carrascosa, María L. Sánchez-Ferrer, Ernesto de la Cruz-Sánchez, Julián J. Arense-Gonzalo, María T. Prieto-Sánchez, Emilia Alfosea-Marhuenda, Miguel A. Iniesta, Jaime Mendiola, Alberto M. Torres-Cantero

**Affiliations:** 1Department of Obstetrics & Gynecology, “Virgen de la Arrixaca” University Clinical Hospital, EI Palmar, 30120 Murcia, Spain; inmagc92@gmail.com (I.G.-C.); marisasanchez@um.es (M.L.S.-F.); mt.prieto@um.es (M.T.P.-S.); emilia.alfosea@hotmail.es (E.A.-M.); mikel.iniesta.albaladejo@gmail.com (M.A.I.); 2Institute for Biomedical Research of Murcia, IMIB-Arrixaca, El Palmar, 30120 Murcia, Spain; jaime.mendiola@um.es (J.M.); amtorres@um.es (A.M.T.-C.); 3Division of Preventive Medicine and Public Health, Department of Physical Activity, Faculty of Sport Sciences, University of Murcia, 30100 Murcia, Spain; erneslacruz@um.es; 4Division of Preventive Medicine and Public Health, Department of Public Health Sciences, University of Murcia School of Medicine, 30100 Murcia, Spain

**Keywords:** skinfold thickness, pregnancy, anthropometric measures, obesity, fat mas, body composition

## Abstract

Anthropometric assessment during pregnancy is a widely used, low-technology procedure that has not been rigorously evaluated. Our objective is to investigate fat mass distribution during pregnancy by examining changes in anthropometrics measures, in order to evaluate the reliability of these measures. An observational, longitudinal, prospective cohort study was performed in 208 pregnant women. Anthropometric measurements were taken following the ISAK protocol during the three trimesters and a generalized linear model for repeated measures was used to evaluate differences. Variability was assessed using the coefficient of variation, and Propagated Error (PE) was used to sum of skinfold thicknesses (SFT). SFT showed a general increase in fat mass during the three trimesters of pregnancy (∑SFT7 *p* = 0.003), and was observed in specific anatomical locations as well: arms (∑Arm SFT, *p* = 0.046), trunk (∑Trunk SFT, *p* = 0.019), legs (∑Leg SFT, *p* = 0.001) and appendicular (∑Appendicular SFT, *p* = 0.001). Anthropometric measures for skinfold thickness were taken individually during pregnancy and were reliable and reproducible during the three trimesters, which could help to prevent adverse pregnancy outcomes.

## 1. Introduction

Anthropometric evaluation allows the estimation of body composition and proportionality in relation to nutrition and growth. Nowadays, anthropometric assessment is widely used by experts from different fields (sports, education, health, engineering, ergonomics) and the quality of their measures determines correct interventions. Indeed, it is important to investigate the ability to measure the success of an intervention, because an ineffective intervention is not likely to be improved otherwise [1].

Anthropometric assessment is a widely used, low-technology procedure that has rarely been rigorously evaluated during pregnancy [2]. There are several anthropometric measures that have been used during preconception and pregnancy to evaluate maternal body composition and changes throughout pregnancy, such as body mass index (BMI; most often pre-pregnancy), gestational weight gain (GWG), and skinfold thickness measurements (SFT). A recent meta-analysis about maternal anthropometry and pregnancy outcomes conducted by the WHO confirmed the inherent value of maternal weight, height, arm circumference and BMI as a predictors of specific infant and maternal outcomes [3]. For instance, GWG has been related to fetal growth and new-born weight [4,5]. Recent studies have pointed out that GWG—both above or below recommendations—is related to a higher risk of intrauterine growth retardation (IUGR), low birth weight (LBW) and prematurity [6]. Currently, recommendations for GWG are based on pre-pregnancy BMI, which shows a positive correlation with birth weight [7].

Multiple investigations have shown that these anthropometric measures are useful but incomplete. GWG and BMI have been proposed as screening methods for identifying pregnancies with abnormal progression that might be at risk of adverse perinatal outcomes. Nonetheless, they provide very limited information regarding changes in women’s body compositions throughout their pregnancies. Moreover, fetal growth may be influenced by more specific maternal tissue changes than by total GWG or BMI [8].

With this in mind, it is important to assess anthropometric measures accurately by evaluating changes in body composition during pregnancy. In this way, SFT has shown a high correlation with the percentage of body fat obtained through other techniques (densitometry, DEXA or dilutional methods) [9]. Calipers are used to measure the thickness of the skin and fat mass at a various points around the body. Following this approach, investigators have used either individual SFTs (the 8 SFT recommended by ISAK) and a sum of several different skinfolds, trying to evaluate different anatomical places (total body fat, trunk, arms, legs, and appendicular zone) joining the different individual SFTs [10,11,12,13]. Thus, this method is useful to describe normal body fat changes throughout gestation, to identify women with unusually small or large changes in body fat during pregnancy, and to estimate initial body fat content [8]. Additionally, using SFT for assessing body composition is a quick, convenient, relatively inexpensive method across all countries, even in low- and middle-income countries. For that reason, it is considered the gold standard among anthropometric measurements [14].

The importance of correctly assessing increases in fat mass throughout pregnancy is essential, due to the dramatic increase of prevalence rates of overweight and obesity among women of childbearing years, and the consequences this has on the offspring [15]. Although obesity is known to negatively affect pregnancy, obesity during pregnancy is commonly ignored [16]. Moreover, maternal obesity increases the risks of hypertensive disorders of pregnancy, gestational diabetes, fetal macrosomia, cesarean deliveries, congenital anomalies and stillbirth [17].

Determination of variability in basic measures and body composition estimates is essential to increasing measurement precision and the reliability of examiners carrying out those measurements [18]. Investigating interventions during pregnancy using unreliable measures may attenuate or overestimate observed associations and make it difficult to detect true etiologic associations. To the best of our knowledge, no data exist concerning the reliability of anthropometric measurements during pregnancy, according to the International Society of the Advancement of Kinanthropometry (ISAK). Therefore, our objective is to investigate fat mass distribution during pregnancy by examining changes in anthropometrics measures, in order to assess this affordable and inexpensive method and evaluate the reliability of these measurements to improve adequate lifestyle interventions, even in low- and middle-income countries.

## 2. Materials and Methods

### 2.1. Study Population

An observational, longitudinal, prospective cohort study was performed in a tertiary Gynaecology and Obstetrics Service of a University Clinical Hospital in Murcia Region (Spain). Recruitment was done from March 2016 to September 2019. Subjects were Caucasian singleton pregnant women attending their routine first trimester ultrasound at the hospital at 11 + 0 to 13 + 6 weeks of gestation. Inclusion criteria were: ≥16 years of age, intention to deliver at the reference hospital, no communication problems and singleton pregnancy. Two hundred and eighty women were invited to participate in the study, of whom 208 were finally accepted (participation rate: 74.3%). Those who refused to participate (*n* = 72) mainly claimed lack of time (*n* = 62), other follow-ups in private hospitals (*n* = 3), not being interested in the research (*n* = 5), or lack of transport to reach the hospital (*n* = 2). Finally, 3 women withdrew in the second trimester and 3 more withdrew in the third, leaving the final sample at 202 women.

Participants signed an informed consent to participate in the study. The study was approved by the hospital Clinical Research Ethics Committee on April 2017 (No 04/17).

### 2.2. Data Collection

Data were collected throughout the three trimesters of pregnancy, 1st visit (11–14 weeks gestation), 2nd visit (16–20 weeks’ gestation) and 3rd visit (28–34 weeks’ gestation). We recorded maternal age, parity, tobacco and alcohol consumption, previous maternal diseases (hypertension or diabetes) and nutritional supplements consumption.

### 2.3. Body Composition and Anthropometric Assessment

Anthropometric measurements were taken following The International Society of the Advancement of Kinanthropometry (ISAK) protocol [19]. Body measurements were collected by three experienced clinicians with anthropometric training (a minimum level 1 by the ISAK). The improved anthropometric assessment standards, based on already established methodologies, as well as an international accreditation scheme utilizing the concept of a four-tier hierarchy, were developed with high rigor and quality maintenance to be ISAK’s differentials, recognized worldwide. A level 1 anthropometrist (Technician—Restricted Profile) comprises a narrow measurement profile and was designed for most ISAK-accredited anthropometrists, who have an ongoing need for more-advanced skinfold measurements [20].

Weight and height were measured using a digital scale (Tanita SC-330S, Amsterdam, The Netherlands) and BMI was calculated.

The adverse metabolic consequences of obesity are related to the accumulation of subcutaneous fat. Thus, subcutaneous fat may be better indicator of adiposity of pregnant women than body weight or BMI [21]. Eight skinfolds (triceps, subscapular, biceps, iliac crest, supraspinal, abdominal, mid-thigh and calf) were taken (mm) by a Slim Guide skinfold caliper following ISAK protocol [19], with a precision of 0.5 mm. All measurements were taken in triplicate on the participant’s right side, and the median values were used.

The sum of skinfolds, a proxy for overall subcutaneous fat, was calculated by summarizing the eight skinfold sites (triceps, subscapular and suprailiac sites) following ISAK protocol and literature [19,21]. The following sums were considered for fat content calculations at different anatomical places: eight skinfolds (∑SFT8) (triceps, subscapular, biceps, iliac crest, supraspinal, abdominal, mid-thigh, calf), seven skinfolds (∑SFT7) (minus iliac crest), upper limb skinfolds (∑Arm SFT) (biceps, triceps), trunk skinfolds (∑Trunk SFT) (subscapular, iliac crest, supraspinal, abdominal) and lower limb skinfolds (∑Leg SFT) (mid-thigh, calf). The suprailiac and subscapular skinfolds, and in general trunk skinfolds, are good predictors of glycaemia and insulin resistance. Other complex measurements such as MRI, DXA and CT make only a small addition to the prediction. This finding supports the application of anthropometry for determining trunk fat, with individual skinfolds or summatory, in clinical and epidemiological settings [21]. Furthermore, upper and lower limb skinfolds and appendicular skinfolds refers to peripheric fat mass in pregnancy [22]. Bone breadth (wrist, humerus and femur) were measured with a Holtain pachymeter with a precision of 1 mm, and girths (cm; including arm girths (relaxed and flexed), waist girth, hip girth, mid-thigh girth and calf girth) were measured with a narrow, metallic and inextensible Rosscraft measuring tape with a precision of 1 mm.

### 2.4. Statistical Analysis

The descriptive statistics of pregnant women are shown as mean (standard deviation, SD), and *n* (%) where appropriate.

We assessed the variability for every single anthropometric measurement in each trimester using the coefficient of variation (CV) for each of the three measurements. The CV is useful in comparing the variability of several different samples [23]. It is a relative measure of variability that allows the variability between disparate groups and characteristics to be compared. The higher the CV, the greater the variability.

For skinfold sums we used the Propagated Error (PE) obtained by
$$\mathrm{PE}=\frac{\sqrt{\sum Var\left(Skinfold_{i}\right)}}{\sum mean\left(Skinfold_{i}\right)}$$

We considered PE as a varying quantification of the skinfold sums, in order to take into account the variability of the sum of several measurements, since uncertainty or variability propagate when a sum of these measurements is considered. As we assessed variability using the CV, the PE was obtained via the CV of the sum of each of the skinfold measurements. Data were summarized in a violin plot.

A generalized linear model (GLM) for repeated measures was used to evaluate differences in the anthropometric measurements within the three trimesters of pregnancy. The GLM procedure included the Wilks’ Lambda or Greenhouse–Geiser test (after checking sphericity using Mauchly’s Test) to assess differences through time. Bonferroni’s pairwise comparison was also used to compare anthropometric measurements among trimesters. We performed a model for each anthropometric measurement as an independent variable, considering the trimester as a factor.

All tests were two-tailed, and the level of statistical significance was set at 0.05. Statistical analyses were performed with the IBM Statistical Package for Social Sciences (SPSS) v25 (IBM Corporation, Armonk, NY, USA).

## 3. Results

The mean age of our patients was 31.9 years (4.7 years), and 87 women (41.8%) were primiparous. Only 2.4% of the patients declared a punctual consumption of alcohol during pregnancy, while 12.5% were smokers, with an average consumption of 4.7 cigarettes/day. As regards women taking supplements during pregnancy, only 1% refused to take any type of supplement. Fifty-seven percent of patients received supplements with folic acid and iodine, while 33% reported taking multivitamin-type supplements, which included vitamin D (5–10 µg/day). Regarding pre-pregnancy BMI, 54% of women were in a normal weight range (BMI 18.5–24.9 kg/m^2^), while 3.4% were underweight and 29.3% and 13.5% were overweight and obese, respectively. The mean GWG was 7.5 kg. Regarding the mode of delivery, 79% of the patients had a vaginal delivery, on which 55% was eutocic and 24% was through instrumentation. Twenty-one percent of the women underwent a caesarean section (Table 1). We have added a Appendix A in order to display weight gain by trimester and by fetal sex.

The triceps, subscapular, iliac crest and mid-thigh SFT increased significantly across the three trimesters of pregnancy: 0.84, 2.72, 5.73 and 2.91 mm, respectively. In contrast, biceps, supraspinal and calf SFT did not show differences across the pregnancy. Fat mass distribution using the sum of skinfolds showed a general increase during the three trimesters of pregnancy (∑SFT7 *p* = 0.003). Moreover, we observed an increase in fat mass in specific parts during pregnancy: arms (∑Arm SFT, *p* = 0.046), trunk (∑Trunk SFT, *p* = 0.019), legs (∑Leg SFT, *p* = 0.001) and appendicular (∑Appendicular SFT, *p* = 0.001).

Regarding other anthropometric measures, girths showed an important increase in all measurements (arm girths (relaxed and flexed), waist girth, hip girth, mid-thigh girth and calf girth, *p* < 0.001). Finally, bone breadths were analyzed. Only femur breadth showed an increase during pregnancy (Table 2).

Reliability and agreement of skinfold, circumferences and diameter measurements for the three trimesters of pregnancy are shown in Table 4, and the Figure 1 shows the density of the propagated error (PE) in sums of SFT among different trimesters, with the ∑Trunk SF being the most affected measure in the last trimester of pregnancy.

Moreover, we found that CV measurements differed for the first trimester of pregnancy only in the mid-thigh skinfold (*p* = 0.027; Table 4). These differences in CV during the first trimester persisted when calculating the PE for different skinfolds sums, including the mid-thigh skinfold (PE of ∑Leg SFT (*p* = 0.001), PE of ∑Appendicular SFT (*p* < 0.000) and PE of ∑SFT7 (*p* < 0.000)), suggesting differences in the consistency of measurements during pregnancy. Despite this, we did not find differences for the iliac crest, supraspinal and abdominal skinfold CVs during the pregnancy period, and the PE ∑Trunk CVs, which contain all these skinfolds, differed across all trimesters (*p* < 0.000). The measurements of girths and bone breadths showed CV differences (*p* < 0.05), excepting for the flexed arm girth and humerus breadth (Table 4).

## 4. Discussion

In our study on singleton pregnant women, we evaluated changes throughout pregnancy in fat mass using the ISAK protocol, and analyzed the reliability and interobserver reproducibility in the anthropometric measurements. Overall, we observed an increase in fat mass due to a significant increase in almost all maternal SFT measurements throughout pregnancy. Moreover, the analyses from CV and PE showed that anthropometric measurements of SFT are reliable during the three trimesters if performed by trained examiners (see Figure 1).

The importance of correctly assessing increases in fat mass during pregnancy may result in a dramatic increase in the prevalence rates of obesity between women of childbearing years and the consequences on their offspring [15]. Regarding pre-pregnancy BMI, 29.3% and 13.5% participants in our study were overweight or obese, respectively. Overweight before and during pregnancy is a public health threat, since it may lead to pregnancy and birth-related complications such as gestational diabetes mellitus, pregnancy-induced hypertension, caesarean section, weight-related issues in infants (in the short term) and chronic diseases (in the long run). Further, maternal obesity may lead to inter-generational cycles of obesity and metabolic syndrome through fetal programming (13).

Commonly used methods to assess body composition include anthropometry, densitometry (air displacement plethysmography, underwater weighing), and hydrometry (isotope dilution, bioimpedance analysis). DXA can also measure total and regionally specific fat mass, and is now the most commonly used method by which to measure total body composition in non-pregnant individuals [24]. However, DXA scanners emit radiation and, because of that, the procedure is considered harmful to a developing fetus and is forbidden during pregnancy. During pregnancy, whole-body imaging can be accomplished via MRI. Sohlström and Forsum were one of the first groups to evaluate whole-body composition by MRI in pregnant women [25]. MRI is considered safe in pregnancy; however, non-diagnostic scanning protocols are indicated only from the second trimester onwards, and use of MRI is limited by is its cost. The BIA and isotope dilution (D_2_O) methods offer the ability to evaluate body composition in fat mass and fat-free mass, but without consideration of the changes in fat-free mass hydration they are prone to errors.

In emerging countries, pregnant women usually start their prenatal care after the first trimester of pregnancy; thus, pre-pregnancy weight can be undefined. In such cases, anthropometric evaluation and total weight gain are difficult to determine [26]. In this way, SFT could be an affordable and inexpensive method that is easy to use and can provide adequate lifestyle interventions to prevent disease even in low- and middle-income countries. The measurement of SFT in pregnancy has been previously reported [2,12,27,28].

Anthropometric measures (including SFT, maternal weight and circumference) have been used to create prediction equations by which to estimate body fat percentage during pregnancy [2,29]. However, most of these are outdated, and their validation studies do not include different kind of morphotypes (i.e., obese women), so the applicability of SFT equations to calculate body composition in women with obesity or overweight is unknown. Many studies have been recently published on this topic [13,26,30,31], although many of these do not explain a specific protocol for measuring skinfolds [2,26,30,31] or, in other cases [12,13,27], SFT is measured following older protocols (i.e., Lohman et al. (1988) [32] or Taggart et al. (1967) [33]). Lohman et al. emphasize that the SFTs that better correlate with body density are the abdominal, triceps, and calf SFT. Therefore, current protocols followed by ISAK [19] typically include measurements at four to eight sites and the simple summation of skinfold thickness measurements at particular areas of the body, which are used to approximate total body subcutaneous fat. Moreover, every measure is explained carefully in order to minimize variability and improve reproducibility.

In this way, we confirm that an accurate protocol for anthropometric measurement during pregnancy is needed, in order to standardize this method. Furthermore, no study has analyzed the difficulties and limitations of measuring these anthropometric variables in pregnancy and whether these measures are reliable or reproducible.

Some studies have reported longitudinal changes in SFT in several populations of pregnant women [12,25,26,29,34]. These results suggest highly variable changes in SFT across measurement sites throughout pregnancy. In our study, we observed an increase in the triceps (0.84 mm), subscapular (2.72 mm), iliac crest (5.73 mm) and mid-thigh (2.91 mm) SFT during pregnancy from 11 to 30 weeks. Similar findings in the range of 1.1–1.9 mm in tricipital have been found by other researchers (Paxton et al. [29], Sidebottom et al. [12] and Mahaba et al. [34]); however, the average subscapular skinfold increase in our research (2.72 mm) was smaller than those found by Sidebottom et al. (4.2 mm), López et al. (4.2 mm) and Forsum et al. (5.9 mm) [25]. In general, differences could be explained by the socioeconomic and demographic characteristics of the populations, and because measurements were performed at different gestational ages.

In all five studies mentioned [12,25,26,29,34], the largest increase was observed in subscapular skinfold. Iliac crest (the largest increase in our study, with 5.73 mm), a new skinfold included in recent protocols, was not evaluated. These findings corroborate the presence of increased central SFT more than those at appendicular and peripheral locations. In addition, this pattern of changes shows a peak increase at the end of the last trimester.

Soltani & Fraser [27] presumed that SFT increase during pregnancy occurred according to a woman’s pre-pregnancy BMI. Their findings show a different pattern of SFT variation for overweight and obese women in comparison to normal-weight women. In our study, it is shown that overweight or obese women in the beginning of pregnancy had lower increases in all measurements of sums of SFT compared with women with a normal BMI. These results concur with those of a previous study, in which women with lower pre-pregnancy BMIs experienced faster gains in SFT during early pregnancy. This study reveals that, in comparison with normal-weight women, obese women had slower rates of SFT gain at the triceps, suprailiac and thigh sites during late pregnancy, as well as smaller total increases in subscapula and suprailiac SFT during pregnancy [13].

Moreover, although the importance of measuring SFT in pregnancy has been evaluated, there are well-documented limitations, many of which are apt to be accentuated by pregnancy. First, SFT measures are influenced by the compressibility of the subcutaneous adipose tissue layer, which can be affected by site, gender, age, recent weight changes and even pregnancy. There is some evidence that skinfold compressibility gradually increases in the second and third trimesters [35], and that SFT measured by calipers overestimates subcutaneous fat in pregnancy compared with other procedures, such as MRI or ultrasound [25,36]. Second, edema that often occurs in pregnancy may also affect the ability to obtain accurate measurements, especially in the leg region [35]. Third, skinfold assessment is quite subjective, and consequently extensive training is required to ensure a high level of reliability between pregnant women. In our study, our investigators had an accreditation scheme by ISAK with the objective of maintaining the quality of measurements. Finally, pregnancy itself presents some particularities that can affect the reproducibility of the method. For example, SFT during pregnancy is often greater in underweight compared to overweight women, greater in primiparous compared to multiparous women, and the increase in thickness differs by site throughout gestation [37]. Additionally, as pregnancy progresses, it becomes difficult to obtain skinfold measurements from the trunk region [38], as we have confirmed. A greater difficulty and lower reliability have been described in the estimation of SFT for women who are overweight or obese, which may explain our findings during the third trimester of pregnancy [39].

Although the use of SFT is useful and easy, we conclude that it is a technique with some limitations during pregnancy. We do not recommend the performance of anthropometric measures with SFT if a qualified person is not available, and recommend not using the sum of SFTs in pregnancy, especially ∑Trunk SFT in the third trimester (Figure 1).

Withstanding these limitations, when precision is preserved, repeated evaluation of SFT during pregnancy can be useful and efficient in both research and clinical settings, even in low- and middle-income countries [37].

In this way, it is fundamental to continue investigating how to measure obesity and fat mass distribution during pregnancy by examining changes in anthropometric measures and looking for more reliable and reproducible techniques, if any.

## 5. Conclusions

To our knowledge, there are no data regarding the analysis and reliability of anthropometric measurements during pregnancy according to the ISAK. Previous studies using anthropometric measures in pregnancy might have diluted or attenuated the detected associations of exposure with the disease of interest. In this way, we conclude that anthropometric measures for SFT taken during pregnancy are reliable and reproducible throughout the three trimesters (except mid-thigh SFT). The simple technique and low costs associated with measurements can enable their implementation by frontline health workers, including in remote or rural areas.

Nonetheless, the sum of SFTs in pregnancy should be used carefully because of the PE, especially the ∑Trunk SFT in the third trimester, and although the use of SFT is useful and easy, it is a technique with some limitations in pregnancy. Consequently, further studies will be necessary to confirm or replicate our findings.

## Figures and Tables

**Figure 1 jcm-10-03933-f001:**
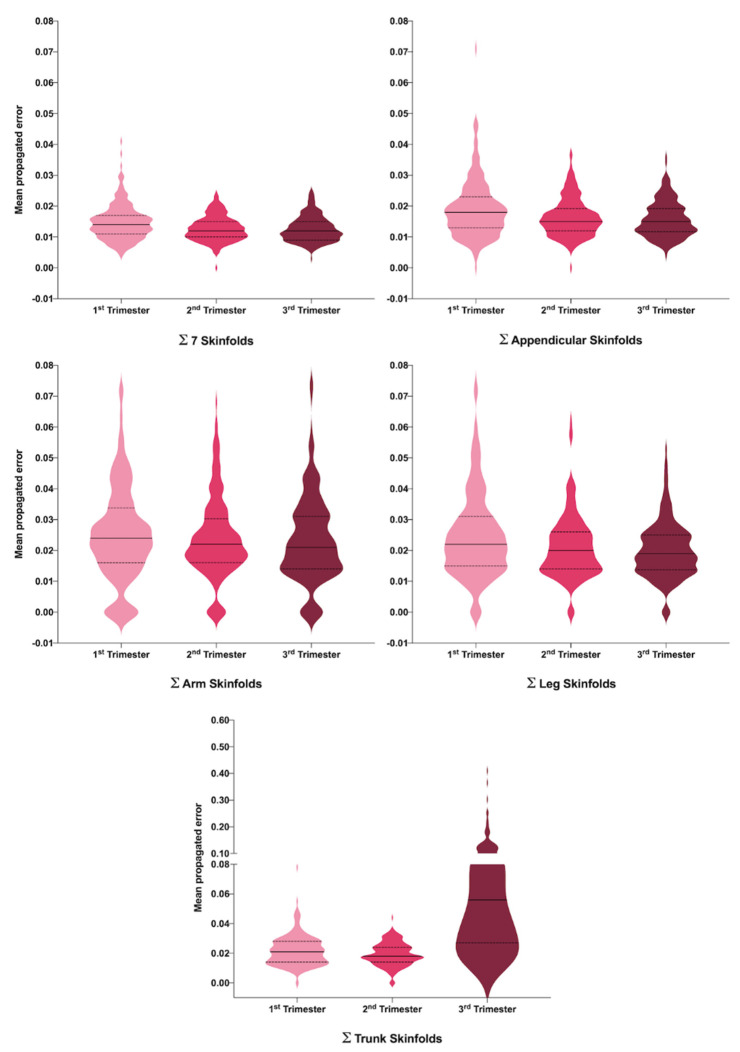
Reliability and agreement of sum skinfolds for the three trimesters during pregnancy represented by violin plots. Anthropometric: ∑7SFT (mm). Sum of seven skinfolds [triceps + subscapular + biceps + iliac crest + abdominal + mid-thigh + calf (mm)]; ∑Appendicular SFT (mm). Sum of appendicular skinfolds [triceps + biceps + mid-thigh + calf (mm); ∑Arm SFT (mm). Sum of arm skinfolds [triceps + biceps (mm)]; ∑Leg SFT (mm). Sum of leg skinfolds [mid-thigh + calf (mm)]; ∑Trunk SFT (mm). Sum of trunk skinfolds [subscapular + iliac crest + abdominal (mm)].

**Table 1 jcm-10-03933-t001:** General characteristics of pregnant women (*n* = 208).

Age (years), mean (SD)	31.93 (4.66)
Pre-pregnancy BMI, *n* (%)	
Underweight (BMI < 18.5 kg/m^2^)	7 (3.4)
Normal weight (BMI 18.5–25 kg/m^2^)	112 (53.8)
Overweight (BMI 25–30 kg/m^2^)	61 (29.3)
Obese (BMI > 30 kg/m^2^)	28 (13.5)
Weight gain (kg), mean (SD)	7.48 (3.4)
Parity, *n* (%)	
Primiparous	87 (41.8)
Multiparous	121 (58.2)
Alcohol consumption, *n* (%) ^a^	5 (2.4)
Tobacco consumption, *n* (%) ^b^	26 (12.5)
Delivery, *n* (%)	
Eutocic	112 (55.4)
Instrumental	48 (23.8)
Caesarean	42 (20.8)

^a^ Did you ever drink alcoholic beverages with a frequency of at least one a month? ^b^ Have you ever smoked?

**Table 2 jcm-10-03933-t002:** Differences in skinfold, circumference, diameter and BMI for the three trimesters during pregnancy.

	1st Trimester	2nd Trimester	3rd Trimester	*p*-Value	Bonferroni’s Post-Hoc Test
Body mass (kg)	67.34 (11.99)	69.42 (11.94)	74.61 (11.96)	0.000	All
BMI	24.78 (4.57)	25.52 (4.50)	27.49 (4.62)	0.000	All
**Skinfolds (mm)**					
Triceps	24.87 (8.50)	25.26 (8.17)	25.71 (7.67)	0.036	T1 vs. T3
Subscapular	19.21 (8.37)	19.90 (8.56)	21.93 (9.41)	0.000	All
Biceps	14.54 (6.91)	14.45 (7.21)	14.91 (6.69)	0.228	-
Iliac crest	23.45 (9.71)	26.29 (9.86)	29.18 (11.14)	0.000	All
Supraspinal	20.62 (9.49)	21.45 (9.39)	21.64 (10.58)	0.262	-
Abdominal	30.61 (12.27)	31.79 (10.47)	30.40 (10.27)	0.032	T1 vs. T2 T2 vs. T3
Mid-thigh	34.03 (9.74)	36.07 (10.21)	36.94 (10.88)	0.000	T1 vs. T2 T1 vs. T3
Calf	24.11 (7.91)	24.00 (8.52)	24.55 (7.94)	0.409	-
∑SFT7	168.16 (53.42)	172.91 (51.83)	176.46 (53.62)	0.003	T1 vs. T2 T1 vs. T3
∑Arm SFT	39.42 (14.6)	39.71 (14.58)	40.62 (13.54)	0.046	T1 vs. T3
∑Trunk SFT	70.43 (27.36)	73.14 (25.54)	74.18 (27.14)	0.019	T1 vs. T2 T1 vs. T3
∑Leg SFT	58.04 (16.38)	60.07 (17.60)	61.54 (17.66)	0.001	T1 vs. T2 T1 vs. T3
∑Appendicular SFT	97.54 (28.51)	99.78 (28.87)	102.04 (28.74)	0.001	All
**Girths (cm)**					
Arm Girth (relaxed)	27.73 (4.05)	28.18 (3.77)	28.54 (3.74)	0.000	All
Arm Girth (flexed)	28.92 (4.20)	29.32 (3.70)	29.56(3.74)	0.000	All
Waist Girth	80.14 (11)	82.97 (10.61)	87.91 (10.11)	0.000	All
Hip Girth	102.30 (10.87)	103.61 (9.59)	106.41 (8.71)	0.000	All
Calf Girth	35.6 (3.15)	35.83 (3.37)	36.54 (3.37)	0.000	All
**Bone breadth (cm)**					
Humerus	8.94 (7.13)	8.33 (4.85)	7.85 (0.79)	0.063	-
Femur	12 (1.368)	11.69 (1.34)	11.97 (1.37)	0.005	T1 vs. T2 T2 vs. T3

Results expressed as mean (standard deviation). GLM for repeated measures with multiple comparisons were performed. Results expressed as T1 vs. T2, T2 vs. T3, T1 vs. T3 or *All* (significant relation between all trimesters). Anthropometric: ∑7SFT (mm). Sum of seven skinfolds [triceps + subscapular + biceps + iliac crest + abdominal + mid-thigh + calf (mm)]; ∑Appendicular SFT (mm). Sum of appendicular skinfolds [triceps + biceps + mid-thigh + calf (mm); ∑Arm SFT (mm). Sum of arm skinfolds [triceps + biceps (mm)]; ∑Leg SFT (mm). Sum of leg skinfolds [mid-thigh + calf (mm)]; ∑Trunk SFT (mm). Sum of trunk skinfolds [subscapular + iliac crest + abdominal (mm)]. We have described changes in SFT stratified by BMI in Table 3. It is shown that overweight or obese women at the start of pregnancy had lower increases in all measurements of sums of SFT compared with women with a normal BMI.

**Table 3 jcm-10-03933-t003:** Differences in skinfold for the three trimesters during pregnancy, stratified by BMI.

	1st Trimester	2nd Trimester	3rd Trimester	*p*-Value	Bonferroni’s Post-Hoc Test
**Normal weight (BMI 18.5–25** **kg/m^2^** **)**					
∑SFT7	135.72 (32.67)	143.43 (31.70)	148.43 (35.03)	0.000	All
∑Appendicular SFT	80.38 (17.96)	84.06 (18.61)	87.40 (20.05)	0.000	All
∑Arm SFT	31.12 (8.87)	32.25 (8.92)	33.96 (9.42)	0.003	T2 vs. T3, T1 vs. T3
∑Leg SFT	49.32 (11.69)	51.72 (12.97)	53.51 (13.41)	0.002	T1 vs. T2, T1 vs. T3
∑Trunk SFT	55.91 (18.35)	59.80 (16.81)	61.04 (17.91)	0.005	T1 vs. T2, T1 vs. T3
**Overweight (BMI 25–30** **kg/m^2^** **)**					
∑SFT7	194.91 (33.33)	198.81 (31.59)	200.74 (38.85)	0.544	
∑Appendicular SFT	112.04 (20.29)	112.98 (18.71)	114.53 (19.92)	0.611	
∑Arm SFT	45.62 (10.46)	45.47 (9.91)	47.23 (10.53)	0.451	
∑Leg SFT	66.42 (14.55)	67.51 (14.25)	67.30 (14.08)	0.856	
∑Trunk SFT	82.87 (17.90)	85.83 (17.64)	86.21 (22.31)	0.485	
**Obese (BMI > 30 ** **kg/m^2^** **)**					
∑SFT7	240.84 (40.86)	246.05 (45.01)	247.95 (56.63)	0.881	
∑Appendicular SFT	134.07 (22.36)	138.00 (27.35)	139.68 (29.58)	0.712	
∑Arm SFT	59.05 (11.35)	61.32 (13.78)	57.50 (12.36)	0.358	
∑Leg SFT	75.02 (14.91)	76.68 (19.78)	82.18 (19.74)	0.118	
∑Trunk SFT	106.77 (23.23)	108.05 (20.11)	108.27 (29.91)	0.941	

Results expressed as mean (standard deviation). GLM for repeated measures with multiple comparisons were performed. Results expressed as T1 vs. T2, T2 vs. T3, T1 vs. T3 or *All* (significant relation between all trimesters). Anthropometric: ∑7SFT (mm). Sum of seven skinfolds [triceps + subscapular + biceps + iliac crest + abdominal + mid-thigh + calf (mm)]; ∑Appendicular SFT (mm). Sum of appendicular skinfolds [triceps + biceps + mid-thigh + calf (mm); ∑Arm SFT (mm). Sum of arm skinfolds [triceps + biceps (mm)]; ∑Leg SFT (mm). Sum of leg skinfolds [mid-thigh + calf (mm)]; ∑Trunk SFT (mm). Sum of trunk skinfolds [subscapular + iliac crest + abdominal (mm)].

**Table 4 jcm-10-03933-t004:** Reliability and agreement of skinfold, circumference and diameter measurements for the three trimesters during pregnancy.

	1st Trimester	2nd Trimester	3rd Trimester	*p*-Value	Bonferroni’s Post-Hoc Test
**Skinfolds (CV)**					
Triceps	0.026 (0.02)	0.023 (0.018)	0.024 (0.021)	0.307	
Subscapular	0.034 (0.031)	0.032 (0.025)	0.029 (0.022)	0.09	
Biceps	0.042 (0.041)	0.044 (0.036)	0.039 (0.033)	0.139	
Iliac crest	0.035 (0.031)	0.034 (0.025)	0.031 (0.02)	0.199	
Supraspinal	0.033 (0.028)	0.032 (0.023)	0.034 (0.024)	0.677	
Abdominal	0.031 (0.021)	0.028 (0.017)	0.03 (0.021)	0.149	
Mid-thigh	0.029 (0.023)	0.023 (0.015)	0.023 (0.016)	0.027	T1 vs. T2 T1 vs. T3
Calf	0.037 (0.029)	0.035 (0.026)	0.032 (0.02)	0.16	
PE ∑SFT7	0.015 (0.006)	0.013 (0.004)	0.013 (0.004)	0.000	T1 vs. T2 T1 vs. T3
PE ∑Arm SFT	0.025 (0.016)	0.024 (0.013)	0.023 (0.014)	0.379	
PE ∑Trunk SFT	0.022 (0.01)	0.019 (0.008)	0.072 (0.062)	0.000	All
PE ∑Leg SFT	0.025 (0.015)	0.021 (0.01)	0.02 (0.009)	0.001	T1 vs. T2 T1 vs. T3
PE ∑Appendicular SFT	0.019 (0.009)	0.016 (0.006)	0.016 (0.006)	0.000	T1 vs. T2 T1 vs. T3
**Girths (CV)**					
Arm Girth (relaxed)	0.008 (0.008)	0.005 (0.006)	0.006 (0.007)	0.002	T1 vs. T2 T1 vs. T3
Arm Girth (flexed)	0.006 (0.009)	0.006 (0.008)	0.005 (0.006)	0.341	
Waist Girth	0.004 (0.004)	0.003 (0.003)	0.003 (0.003)	0.000	T1 vs. T2 T1 vs. T3
Hip Girth	0.004 (0.006)	0.003 (0.004)	0.002 (0.002)	0.000	All
Calf Girth	0.004 (0.005)	0.004 (0.004)	0.003 (0.004)	0.034	T1 vs. T3
**Bone breadth (CV)**					
Humerus	0.01 (0.017)	0.012 (0.019)	0.008 (0.016)	0.143	
Femur	0.014 (0.015)	0.014 (0.018)	0.008 (0.013)	0.001	T1 vs. T3 T2 vs. T3

Results expressed as mean of coefficient of variation (standard deviation). GLM for repeated measures with multiple comparisons were performed. Results expressed as T1 vs. T2, T2 vs. T3, T1 vs. T3 or *All* (significant relation between all trimesters). Anthropometric: ∑7SFT (mm). Sum of seven skinfolds [triceps + subscapular + biceps + iliac crest + abdominal + mid-thigh + calf (mm)]; ∑Appendicular SFT (mm). Sum of appendicular skinfolds [triceps + biceps + mid-thigh + calf (mm); ∑Arm SFT (mm). Sum of arm skinfolds [triceps + biceps (mm)]; ∑Leg SFT (mm). Sum of leg skinfolds [mid-thigh + calf (mm)]; ∑Trunk SFT (mm). Sum of trunk skinfolds [subscapular + iliac crest + abdominal (mm)].

## Data Availability

The data sets generated and/or analyzed during the current study are available from the corresponding author on reasonable request.

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
