# Peer review of "Analysis and Reliability of Anthropometric Measurements during Pregnancy: A Prospective Cohort Study in 208 Pregnant Women"

_jcm, 2021, doi:10.3390/jcm10173933_

Round 1
Reviewer 1 Report
I am happy with the changes made regarding my previous suggestions. The paper does fill a gap in the research. My only suggestion is to have the paper edited for some minor language errors.
Reviewer 2 Report
Authors have improved the manuscript and clarified accordingly.
This manuscript is a resubmission of an earlier submission. The following is a list of the peer review reports and author responses from that submission.
Round 1
Reviewer 1 Report
Line 52: “the quality of their measures determines the success of their interventions” I would say that it determines their ability to measure the success of their intervention, but it won’t improve an ineffective intervention. Line 58: BMI is not used during pregnancy in my country – it is only used to assess pre-pregnancy body weight. Line 72-73: Is this true for pregnant women as well? Or just non-pregnant. 92: You measured reliability but not validity. While it is worthwhile to look at reliability of this skinfold method, just looking at reliability without also investigating validity or even the effect of factors such as edema makes this study a bit boring. Perhaps rewriting as a short report would be more appropriate with an accurate description of the study's objectives.Author Response
Please see the attachment

Reviewer 2 Report
Overall this paper provides a contribution that the readership of the journal would be interested in. Uses SFT as a proxy for fat mass could be a useful method in resource-poor settings, especially, since pregnancy fat mass is a predictor of a number of infant and maternal perinatal outcomes. However, the authors must do a better job of defining reliability and validity and explaining how their methods test these concepts. The discussion could be more concise and better engage with previous studies about how to interpret the authors’ results.
Introduction
-Could be more specific with line 52. What kind of interventions?
-In the first and second paragraphs the authors use ‘anthropometric evaluation’ and then ‘anthropometric assessment’. Are those different?
-‘many skinfold thickness’ is phrased oddly in line 58
-Need citation for sentence ending on line 62
-Authors might want to be more specific about the sum of skinfolds they reference on lines 69-70. There are several different sums that are used for different purposes, and it would be useful to the reader to have the authors describe those
-The paragraph on skinfolds, lines 66-75, could be better organized. Sentences do not flow (especially the sentence about cost and technical training.
-“Unreliable measurements…” sentence in line 85 is unclear and needs editing.
-Have the authors also thought about how these SFT methods could be used in certain contexts, like low SES hospitals? Would they/could they be used to predict outcomes?
-“ Therefore, our objective is to investigate fat mass distribution during pregnancy by examining changes in anthropometrics measures, in order to evaluate validity and reliability of these measurements to improve adequate lifestyle interventions.” How are the authors assessing validity and reliability? How are the authors defining validity and reliability? Be specific. As I read on, it looks like they only tested reliability.
-The authors could do a better job of clearly laying out the purpose of the paper in the last paragraph of the introduction.
Materials and Methods
Study population
- “Those who refused to participate 103 mainly claimed lack of time.” Can you quantify how many women claimed lack of time? What were some other reasons (with sample sizes)?
Body composition and anthropometric assessment
-This sentence needs grammatical revision: “Technical error in 118 the measurement of skinfolds was less to 5% and to 1.5% in the rest of the measurements “
-I believe the authors should review the relevance of the SFT sums that they used in the introduction. What are these sums associated with in the literature and what is the utility of using different sums? (page 3, lines 124-132)
-Further the authors only have one sentence on the researchers who conducted the SFT. Isn’t the high quality measurement of SFT in this study the basis for their argument that these measures are valid? Perhaps expanding more on the training of the clinicians who conducted the SFT would be beneficial.
Statistical Analysis
-Authors could do a better job of explaining how coefficient of variation and propagated error tests within variability, as this measurement is the basis for the claim of the entire paper.
Results
-line 156: “About supplements during pregnancy,” is oddly phrased
Table 2
- Make sure the table is all commas (there were some periods)
-When reporting the GLM and elsewhere, report more clearly what was in the model. What are the dependent and independent variables?
Discussion
-dramatic, not dramatical, line 221
-the third and fourth paragraphs of the discussion do not seem to fit in the discussion section. Are these more fit for the introduction?
-the fifth paragraph needs revision for grammar and needs to be more detailed in its analysis of other articles written on SFT
-overall there needs to be better and clearer engagement with the previous literature written on this topic and the authors’ results
Reviewer 3 Report
Authors did a good job investigating fat mass distribution during pregnancy using ISAK protocols.
Introduction and methods section are well described and clear.
Results:
- In table 1 can authors describe wight gain by trimester and fetal sex and other neonate descriptions. It is known that excess maternal adiposity can me correlated with adverse birth outcomes as Macrosomia.
- Can authors describe changes in SFT by stratified by BMI. as weight gain recommendation changes depending on pre-gestational BMI. This new analysis can be added as supplementary data if JCM doesn't allow more figures.
Discussion:
- Can authors discuss more MRI/DEXA and dilution methods and the difference in predicting maternal adiposity during pregnancy. As well as more discussion with authors findings.
- Also add the reason why Loghman protocols should be 'replaced" by the ISAK.
Conclusions are good and authors don't overstate their findings.
